# Differential Effects of Sucrase-Isomaltase Mutants on Its Trafficking and Function in Irritable Bowel Syndrome: Similarities to Congenital Sucrase-Isomaltase Deficiency

**DOI:** 10.3390/nu13010009

**Published:** 2020-12-22

**Authors:** Diab M. Husein, Sandra Rizk, Hassan Y. Naim

**Affiliations:** 1Department of Biochemistry, University of Veterinary Medicine Hannover, Bünteweg 17, 30559 Hannover, Germany; diab.husein@tiho-hannover.de; 2Department of Natural Sciences, Lebanese American University, Beirut 1102-2801, Lebanon; sandra.rizk@lau.edu.lb

**Keywords:** sucrase-isomaltase, congenital sucrase-isomaltase deficiency, irritable bowel syndrome, protein trafficking

## Abstract

Congenital sucrase-isomaltase deficiency (CSID) is a rare metabolic intestinal disorder with reduced or absent activity levels of sucrase-isomaltase (SI). Interestingly, the main symptoms of CSID overlap with those in irritable bowel syndrome (IBS), a common functional gastrointestinal disorder with unknown etiology. Recent advances in genetic screening of IBS patients have revealed rare SI gene variants that are associated with IBS. Here, we investigated the biochemical, cellular and functional phenotypes of several of these variants. The data demonstrate that the SI mutants can be categorized into three groups including immature, mature but slowly transported, and finally mature and properly transported but with reduced enzymatic activity. We also identified SI mutant phenotypes that are deficient but generally not as severe as those characterized in CSID patients. The variable effects on the trafficking and function of the mutations analyzed in this study support the view that both CSID and IBS are heterogeneous disorders, the severity of which is likely related to the biochemical phenotypes of the SI mutants as well as the environment and diet of patients. Our study underlines the necessity to screen for SI mutations in IBS patients and to consider enzyme replacement therapy as an appropriate therapy as in CSID.

## 1. Introduction

Digestion of starch, glycogen, sucrose, maltose and other carbohydrates in the intestinal lumen is achieved by the concerted action of a family of microvillar enzymes, the disaccharidases. The digestion of α-glycosidic linkages of carbohydrates commences by salivary and pancreatic α-amylases and is continued in the small intestine by two major mucosal α-glycosidases, sucrase-isomaltase (SI EC 3.2.148 and 3.2.1.10) and maltase-glucoamylase (MGAM¸ EC 3.2.1.20 and 3.2.1.3) [1]. Reduced expression levels or complete absence of intestinal disaccharidases at the cell surface of the enterocytes is associated with carbohydrate maldigestion and malabsorption, most notably described in several cases of genetically-determined sucrase-isomaltase deficiency (CSID) [2,3,4,5,6]. CSID is elicited by single-nucleotide polymorphisms (SNPs) in the coding region of the *SI* gene generating hypomorphic *SI* gene variants (SIGVs) that are distributed over both domains of the SI protein [7]. The main symptoms of CSID are abdominal pain or discomfort, bloating and osmotic often acidic diarrhea [8]. The concept of phenotypic heterogeneity of SI mutants in CSID emerged from biochemical and cellular studies, which led to the classification of the SI mutants into different categories that vary in their intracellular localization, cell surface localization (apical/basolateral), proteolytic processing and function [7,9]. Recently, SIGVs were demonstrated in association with irritable bowel syndrome (IBS) [10,11,12]. IBS is a functional gastrointestinal disorder with an unknown etiology affecting more than 10% of the western population causing abdominal pain and relapsing symptoms of diarrhea (IBS-D), constipation (IBS-C) and other mixed bowel patterns (IBS-M) [13,14]. The discovery of a highly common *SI* gene variant coding for SI, V15F, that results in reduced enzymatic activity, impaired trafficking and reduced cell surface expression of mutant SI, strongly suggested that SI is implicated in the etiology of IBS [10]. This notion has been recently supported by the identification of several SIVGs with higher prevalence in IBS patients as compared to healthy individuals [11]. Biochemical and functional analysis of mutants containing common SIVGs, such as Y975H, R774G and E1414K, confer also increased risk of IBS-D in patients, who are not responsive to the low fermentable oligosaccharides, disaccharides, monosaccharides, and polyols (FODMAPs) diet that does not exclude sucrose [12,15]. In this study we have investigated at the biochemical and cellular levels the trafficking, cell surface expression and function of several rare SIGVs that have been proposed to be pathogenic and are associated with an increased risk for IBS [11]. The data reveal a distinct categorization of these variants based on the intracellular processing and trafficking profiles as well functional capacities of the SI mutants compatible with a heterogeneous pattern of this functional gastrointestinal disorder.

## 2. Materials and Methods

### 2.1. cDNA Construction

*SI* gene variants in IBS were identified in a cohort of 2207 patients by Garcia-Etxebarria et al. [11]. Mutagenesis simulation, plasmid and primer design were performed using SnapGen (GSL Biotech LLC, Chicago, IL, USA). pSG8-SI [16] plasmid was used as a template in the site-directed mutagenesis PCR and the phusion polymerase was used for the amplification of the DNA (Thermo Fischer, Schwerte, Germany). The resulted pSG8-SI mutated plasmids were validated by sequencing (Macrogen, Amsterdam, The Netherlands). The mutations that were introduced into the wild type cDNA are presented in Table 1.

### 2.2. Transfection, Immunoprecipitation, SDS-PAGE and Western Blotting

COS-1 cells grown in 100-mm culture dishes containing DMEM medium were maintained at 37 °C in a humidified atmosphere and 5% CO_2_. For transient transfections, the cells were transfected using the diethylaminoethyl (DEAE)-dextran method as described before [17]. The cells were washed twice with sterile phosphate buffered saline (PBS) (pH 7.4) before solubilization using lysis buffer (25 mM Tris-HCl, pH 7.4, 50 mM NaCl, 0.5% sodium deoxycholate and 0.5% Triton X-100). Wild type and SI mutants were immunoprecipitated as previously described [9]. The immunoprecipitates were subjected to endo-β-N-acetylglucosaminidase H (endo H) and loaded on SDS-PAGE or processed further for activity assays [4,9]. The activities of SI were assessed using the substrates sucrose and palatinose. The specific activities were assessed versus the SI protein bands identified in the Western blot [4,9,12].

SDS-PAGE and Western blottings were performed as described before [4,9,10,12]. Briefly, the immunoprecipitates were mixed with Laemmli loading buffer and DTT and heated for 5 min at 95 °C. The samples were loaded on 6% SDS gels and blotted onto PVDF membranes. The following anti-SI monoclonal antibodies were used to identify the SI proteins: HBB2/614/88 and HBB3/705/60 antibodies [18] followed by HRP-conjugated secondary antibody. The chemiluminescence signals were detected using the ChemiDoc MP™ Touch Imaging System (Bio-Rad, Munich, Germany).

### 2.3. Biotinylation Assay

Transfected COS-1 cells were treated with Biotin (EZ-Link™ Sulfo-NHS-SS-Biotin) [10,12] and the biotinylated cells were processed for immunoprecipitation [4,9]. Each immunoprecipitate was divided in two equal samples that were analysed by SDS-PAGE and transblotted onto PVDF membranes. One blot was developed using streptavidin-HRP that binds the biotinylated SI proteins on the cell surface, whereas the second blot was developed using anti-SI antibodies.

### 2.4. Confocal Fluorescence Microscopy

COS-1 cells were seeded on cover slips and transfected as described in earlier sections with cDNAs encoding wild type SI or SI mutant. The cells were processed for immunofluorescence as described before [9]. For detection of SI, the following antibodies were used: monoclonal mouse anti-SI antibodies (mAb) hSI2,HBB1/691/79, HBB2/614/88, HBB2/219/20, and HBB3/705/60 [18]. For intracellular localization, antibodies against the ER marker calnexin was used in addition to DAPI for nuclear staining. The fluorescence images were visualized in a Leica TCS SP5 confocal microscope with HCX PL APO lambda blue 40.0 × 1.25 OIL UV.

### 2.5. Statistical Analysis

The detection and quantification of the protein bands were accomplished using Image Lab (Bio-Rad Laboratories GmbH, Munich, Germany). The calculations were performed in Microsoft excel. The data represents the results obtained from at least three independent repeats and the reported error bars represent the standard error of the mean (SEM). The statistical analyses, paired or unpaired student *t*-test, were performed using GraphPad Prism with the statistical significance set at * *p* < 0.05, ** *p* < 0.005 and *** *p* < 0.0005.

## 3. Results

SI is normally processed along the secretory pathway from a mannose-rich N-glycosylated form in the endoplasmic reticulum (ER) to a glycosylated mature protein in the Golgi complex prior to sorting in the trans-Golgi network to the apical membrane [4,19]. To assess the trafficking behavior of the SI mutants, we determined their sensitivity towards endo H, which specifically cleaves mannose-rich N-glycans that are a characteristic of ER-located proteins. As shown in Figure 1, the SI mutants harboring the mutations R250C, Y867H and E640G are mostly endo H-resistant, implying that they have been converted to a complex glycoform (245 kDa) and have egressed the ER in a fashion similar to wild type SI. By contrast, the mutations V717D in the isomaltase subunit and P1200S in the sucrase subunit affected the trafficking of SI substantially and resulted in its intracellular block in the ER, as assessed by the complete sensitivity to endo H and deglycosylation of the mannose-rich glycans. Another mutant protein phenotype of SI is elicited by the mutations R1484H and Y1417X, both located in the sucrase domain, which have generated truncated predominantly endo H-sensitive SI mutants. Interestingly, despite these truncations, a partial maturation of these SI mutants to complex glycosylated species in the Golgi apparatus occurs, since faint, but definite endo H-resistant proteins bands could be detected. Finally, the mutation F1625V in sucrase resulted in an intracellular ER-block due to its sensitivity to endo H. Here again, complex glycosylation of the SI mutant can be also observed, although to a minor extent, concomitant with partial or delayed trafficking of this mutant out of the ER.

The biosynthetic features of the SI mutants were further corroborated by cellular localization analyses using confocal laser microscopy. As shown in Figure 2, the transport competent SI mutants containing the mutations R250C, Y867H and E640G co-localize with the red calnexin signal in the ER as well as in fluorescein isothiocyanate (FITC)-labelled structures lining the cell surface. The green signal of SI mutant triggered by F1625V on the cell surface is less intense than the previously described mutants and it co-localizes predominantly with the calnexin in the ER. The SI variants harboring the mutations R1484H, P1200S, Y1417X and V717D co-localize predominantly with the ER-marker calnexin.

The immunofluorescence images support the biosynthetic characteristics of the SI mutants; nevertheless, they do not provide precise data on the level of cell surface expression of the SI mutants, particularly those with a wild type-like maturation pattern. Therefore, a biotinylation approach [10,12] was utilized to measure the cell surface expression of the SI mutants. As shown in Figure 3A we observed substantial variations in the cell surface localization of the mutants that are endo-H resistant, including those containing the mutations Y867H and E640G (56.75 ± 12.05% and 77.6 ± 4.83% respectively), as well as the partially transported SI mutant elicited by F1625V (38.8 ± 6.851%). Thus, despite the wild type-like maturation profiles in the Golgi, these mutations ultimately impact post-Golgi transport.

Finally, we investigated the influence of the mutations on the specific enzymatic activities of SI towards the two substrates sucrose and palatinose. The data presented in Figure 3B reveals significant reductions in the digestive capacity of all the mutants towards palatinose (digested by isomaltase) and sucrose. Whereas a substantial reduction in the sucrase and isomaltase activities (60–90%) was induced by the mutations F1625V, P1200S and R1484H, the ER-blocked SI mutants containing the mutations V717D and Y1417X exhibited a substantial loss or even a complete loss of activity. Finally, the mutations R250C, Y867H and E640G resulted also in reduced enzymatic activities of sucrase and isomaltase, albeit to a lesser extent as compared to the other SI mutants.

## 4. Discussion

The biosynthesis, transport and activity of the SI mutants described here can be categorized into three different groups including immature, mature but slowly transported, and mature and properly transported but with reduced activity. Upon consideration of the impaired trafficking and/or function, we conclude that all the mutants analyzed have reduced carbohydrate digestive capacity at the cell surface. This applies particularly to those mutants with severe phenotypes that are blocked in the ER and exhibit complete loss of activity. We also unraveled SI mutant phenotypes that are deficient but generally not as severe as those associated with CSID patients in our prior investigations [3,4]. In fact, the SI mutants generated by R250C, Y867H and E640G, which are processed in the Golgi to an endo H-resistant species similar to wild type SI, are not trafficked efficiently from the Golgi to the cell surface and additionally reveal variable functional deficits. These two criteria together result in a massive reduction in the carbohydrate digestive capacity of these mutants at the cell surface. Similar mutations have been reported to be associated with a lack of response to a low FODMAP diet due to the reduced enzymatic activities and cell surface expression [12,15] and in several cases of CSID [9]. The severity could be even higher when considering the partially trafficked, ER-located and truncated SI mutants elicited by the mutations F1625V, V717D and R1484H respectively. Together with the low expression at the cell surface, it can be concluded that the mutations have triggered a severe SI phenotype that contributes to the pathogenicity of IBS in individuals harboring these mutations. In general, an association between the severity of the analyzed SIGVs and the diagnosed IBS subtypes [11] is not definitive at this stage of investigation. Nevertheless, the mutations R250C, Y867H and E640G that triggered a mild SI phenotype were found mainly in IBS-D and IBS-C patients and the mutations that elicited delayed SI trafficking, F1625V, or an ER arrest, P1200S, were found only in IBS-D patients. Interestingly, both SI mutations encoding the truncated SI mutants, R1484H and Y1417X, were identified in IBS-U patients.

The variable effects on the trafficking and function of the mutations analyzed in this study support the view that both CSID and IBS associated with hypomorphic SI mutants are heterogeneous disorders, the severity of which is likely related to the biochemical phenotypes of the mutants, as well as the environment and diet of patients. SI exhibits a wide α-glucosidase activity profile and cooperates with another intestinal disaccharidase, maltase-glucoamylase (MGAM) [20,21] in digesting α-1,4 linkages, the major glycosidic linkages in starchy foods. Undoubtedly, it would be intriguing to determine the extent of contribution of MGAM to the activity of SI and vice versa and how each enzyme would compromise the lower expression levels and the reduced digestive function of the other. The gained knowledge could certainly contribute to a better understanding of the molecular background (s) for the variations in the severity levels of the symptoms in IBS individuals.

In essence, this study underlines the role of SI in the etiology of IBS as a primary genetic risk factor in the general population, as well as the necessity of genetic screening of *SI* gene in IBS patients. Further, it recommends a personalized therapeutic approach comprising enzyme replacement therapy or diet regimen that avoid carbohydrates, which are normally digested by SI.

## Figures and Tables

**Figure 1 nutrients-13-00009-f001:**
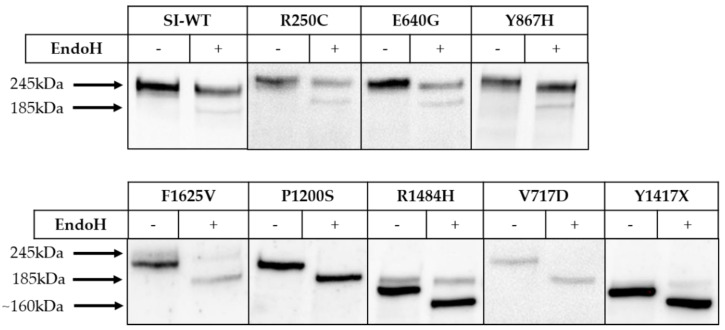
Biosynthetic forms of wild type sucrase-isomaltase (SI) (SI-WT) and SI mutants in COS-1 cells. SI-WT or SI-mutants (R250C, R1484H, E640G, F1625V, P1200S, Y867H, Y1417X and V717D) were expressed in COS-1 cells, immunoprecipitated with mAb anti-SI and treated or not treated with endo H then analyzed by Western blotting. The mature, endo H-resistant form of SI has a molecular weight of 245 kDa; the mannose-rich ER form of SI (210 kDa) is sensitive to endo H and shifts to a lower Band at 185 kDa.

**Figure 2 nutrients-13-00009-f002:**
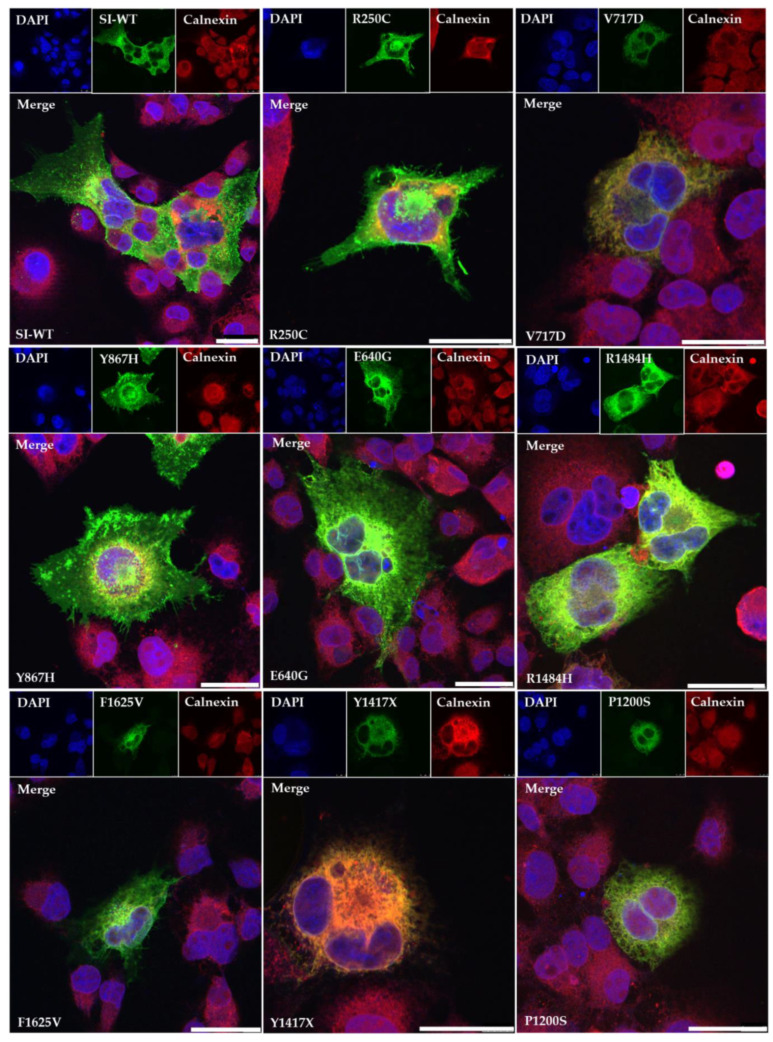
Cellular localization of wild type SI (SI-WT) and SI mutants in COS-1 cells. The SI-WT or Scheme 488. (green). Calnexin was utilized as a protein marker of the ER. Here, rabbit anti-calnexin antibody was used as a primary antibody and goat anti-rabbit IgG (secondary) carrying Alexa flour 568 (red). DAPI labels the nucleus (blue). Bars = 30µm.

**Figure 3 nutrients-13-00009-f003:**
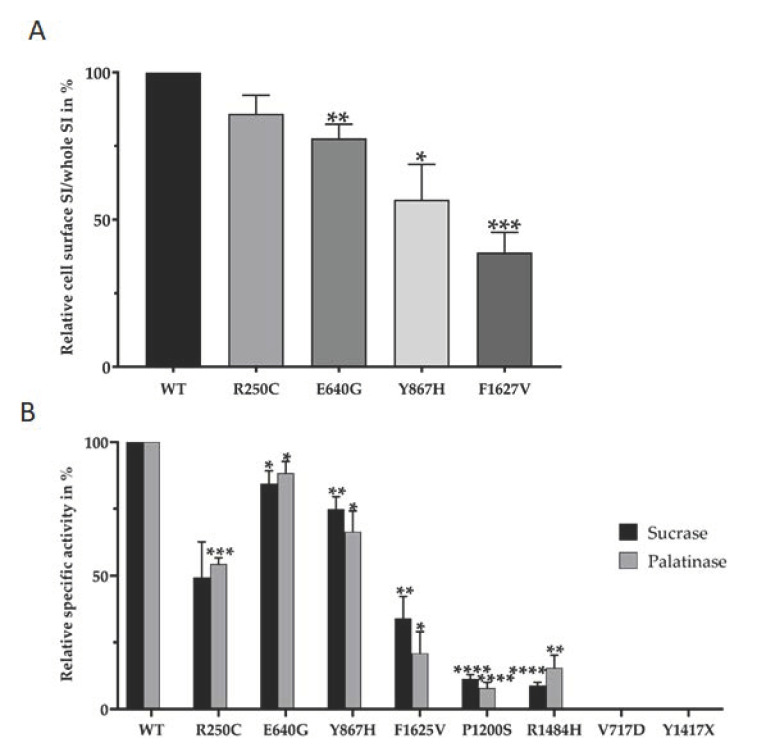
(**A**) Cell surface expression of the SI mutants. SI mutants were expressed in COS-1 cells Table 100. (Unpaired *t*-test, * *p* < 0.05, ** *p* < 0.005 and *** *p* < 0.0005, S.E.M., *n* ≥ 3). (**B**) Relative Specific enzyme activities of the SI mutants. SI-WT or SI mutants were expressed in COS-1 cells and immunoprecipitated by mAb anti-SI. The immunoprecipitates were assayed for enzymatic activity using sucrose or palatinose as substrates. Similar samples were analyzed by Western blotting and the protein band intensity was used to quantify the specific activities of the SI proteins. Relative specific activities are measured in comparison to wild type SI that was set to 100% for sucrase or palatinase. (Paired *t*-test, *******
*p* < 0.05, ** *p* < 0.005, *** *p* < 0.0005 and **** *p* < 0.00005 S.E.M., *n* ≥ 3).

**Table 1 nutrients-13-00009-t001:** Presentation of the mutations in the *SI* gene (see Reference [11]) that have been analyzed in this study, the resulting amino acid exchange and the sequence of the forward oligonucleotide primer used for generating the specific mutations in cDNA encoding *SI*. The modified nucleotide triplet is bold and underlined.

Mutation	Primer	Amino Acid Exchange
c.748C>T	gagattt**tgt**catgatttatcctggaaaacat	Arg→Cyst (p.R250C)
c.1919A>G	tgtggct**ggt**accacagaagaactttg	Glu→Gly (p.E640G)
2150T>A	aagacca**gat**cttcatgagttttatgagg	Val→Asp (p.V717D)
c.2599T>C	attcatca**cat**caggaaggaactaccttag	Tyr→His (p.Y867H)
c.3598C>T	tttgggc**tca**actccacaagttgcaa	Pro→Ser (p.P1200S)
c.4251T>A	acgaacttaat**taa**ccaccttatttcccagaac	Tyr→Stop (p.Y1417X)
c.4451G>A	aatttct**cat**tccacgtatcctactagtgg	Tyr→Hist (p.Y1484H)
c.4873T>G	gggatata**gtc**aagcagttcttatgggg	Phe→Val (p.F1625V)

## Data Availability

Data available in a publicly accessible repository.

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
