# Peer review of "Differential Effects of Sucrase-Isomaltase Mutants on Its Trafficking and Function in Irritable Bowel Syndrome: Similarities to Congenital Sucrase-Isomaltase Deficiency"

_nutrients, 2020, doi:10.3390/nu13010009_

Round 1

Reviewer 1 Report

Certainly innovative work allows a differential diagnosis, but not an adequate therapy that requires pharmacokinetic skills and appropriate pharmacodynamicscertainly innovative work allows a differential diagnosis, but not an adequate therapy that requires pharmacokinetic skills and appropriate pharmacodynamics.

Author Response

Responses to the reviewers

We thank the reviewers for their positive assessment of the manuscript, their suggestions and constructive comments, all of which have been addressed in the revised version of the manuscript. The revisions made are marked.

Reviewer 1

Certainly innovative work allows a differential diagnosis, but not an adequate therapy that requires pharmacokinetic skills and appropriate pharmacodynamics

Response:

The main aim of this study is to investigate hypomorphic SI gene variants (SIGVs) and demonstrate their effects on SI trafficking and function. We could show that these variants represent a risk factor for IBS, albeit at variable levels due to the heterogeneity of the effects elicited by these mutations. The currently available and adequate therapy is an enzymatic replacement therapy (line 242-243 in the revised version), for instance using invertase from S. cerevisiae (or Sucraid) or personalized dietary measures that avoid sucrose and starch rich nutrients. Based on the data from this study and also analyses of additional SI mutations, therapeutic strategies can be designed that could consider pharmacokinetics and pharmacodynamics.

Reviewer 2 Report

It’s a very interesting paper aimed to  investigate  the trafficking, cell surface expression and function of several rare SIGVs that have  been proposed to be associated with an increased risk for IBS. The study could shed new light on  the IBS  pathophysiology

It is not clear at all how many subjects were involved in the study and their clinical features: M, F; mean age…..

-Were all IBS ? and, if so,  which subtypes?

-Did the authors  have a control group?

-Was there any relationship between SIGVs and  symptom severity?

The discussion should, consequently, be modified on the basis of these new items.

Author Response

Responses to the reviewers

We thank the reviewers for their positive assessment of the manuscript, their suggestions and constructive comments, all of which have been addressed in the revised version of the manuscript. The revisions made are marked.

Reviewer 2

  1. It is not clear at all how many subjects were involved in the study and their clinical features: M, F; mean age…

Response:

In this study we have analyzed at the biochemical and cellular levels hypomorphic SI gene variants (SIGVs) that have been initially identified at the genetic level by Garcia-Etxebarria, K. et al. (Increased Prevalence of Rare Sucrase-isomaltase Pathogenic Variants in Irritable  Bowel Syndrome Patients. Clin. Gastroenterol. Hepatol. 2018, 16, 1673–1676, doi:10.1016/j.cgh.2018.01.047) (Ref. # 11 in this manuscript).  The authors of this manuscript did not provide information on the age or gender of the subjects. Although this publication has been referenced in the initial version of the paper, we have additionally included a sentence in the Materials and Methods that refers to this publication (line 64 and also line 71 in Table 1).

  1. Were all IBS? And, if so, which subtypes?

and

  1. Did the authors have a control group?

Response:

According to the paper by Garcia-Etxebarria et al. 2018 (Ref. # 11) all analyzed patients (n=2207) were IBS patients with the following subtypes: IBS-C (n=598), IBS-M (n=503), IBS-D (n=952) and IBS-U (n=154).  The control group was an ethnically matched (non-Finnish, European ancestry; n=33,370) reference population, the frequency of relevant SIGVs were extracted from Exome Aggregation Consortium (ExAC) (http://exac.broadinstitute.org).

  1. Was there any relationship between SIGVs and symptom severity?

Response:

The symptom severity was not described in the study of Garcia-Etxebarria et al. 2018 (Ref. # 11). Nevertheless, we modified the discussion (line 224-229) to address a potential association between the SIGVs and the IBS subtypes.